# Serum Osmolarity and Vasopressin Concentration in Acute Heart Failure—Influence on Clinical Course and Outcome

**DOI:** 10.3390/biomedicines10082034

**Published:** 2022-08-20

**Authors:** Mateusz Guzik, Mateusz Sokolski, Magdalena Hurkacz, Agata Zdanowicz, Gracjan Iwanek, Dominik Marciniak, Robert Zymliński, Piotr Ponikowski, Jan Biegus

**Affiliations:** 1Institute of Heart Diseases, Wroclaw Medical University, 50-556 Wrocław, Poland; 2Department of Clinical Pharmacology, Wroclaw Medical University, 50-556 Wrocław, Poland; 3Department of Drugs Form Technology, Wroclaw Medical University, 50-556 Wrocław, Poland

**Keywords:** acute heart failure, osmolarity, vasopressin, outcomes

## Abstract

Neurohormone activation plays an important role in Acute Heart Failure (AHF) pathophysiology. Serum osmolarity can affect this activation causing vasopressin excretion. The role of serum osmolarity and vasopressin concentration and its interaction remain still unexplored in AHF. The objective of our study was to evaluate the relationship of serum osmolarity with clinical parameters, vasopressin concentration, in-hospital course, and outcomes in AHF patients. The study group consisted of 338 AHF patients (male (76.3%), mean age of 68 ± 13 years) with serum osmolarity calculated by the equation: 1.86 × sodium [mmol/L] + (glucose [mg/dL]/18) + (urea [mg/dL]/2.8) + 9 and divided into osmolarity quartiles marked as: low: <287 mOsm/L, intermediate low: 287–294 mOsm/L, intermediate high: 295–304 mOsm/L, and high: >304 mOsm/L. There was an increasing age gradient in the groups and patients differed in the occurrence of comorbidities and baseline clinical and laboratory parameters. Importantly, analysis revealed that vasopressin presented a linear correlation with osmolarity (r = −0.221, *p* = 0.003) and its concentration decreased with quartiles (61.6 [44.0–81.0] vs. 57.8 [50.0–77.3] vs. 52.7 [43.1–69.2] vs. 45.0 [30.7–60.7] pg/mL, respectively, *p* = 0.034). This association across quartiles was observed among de novo AHF (63.6 [55.3–94.5] vs. 58.0 [50.7–78.6] vs. 52.0 [46.0–58.0] vs. 38.0 [27.0–57.0] pg/mL, respectively, *p* = 0.022) and was not statistically significant in patients with acute decompensated heart failure (ADHF) (59.5 [37.4–80.0] vs. 52.0 [38.0–74.5] vs. 57.0 [38.0–79.0] vs. 50.0 [33.0–84.0] pg/mL, respectively, *p* = 0.849). The worsening of renal function episodes were more frequent in quartiles with higher osmolarity (4 vs. 2 vs. 13 vs. 11%, respectively, *p* = 0.018) and patients that belonged to the quartiles with low and high osmolarity were characterized more often by incidence of worsening heart failure (20 vs. 9 vs. 10 vs. 22%, respectively, *p* = 0.032). There was also a U-shape distribution in relation to one-year mortality (31 vs. 19 vs. 23 vs. 37%, respectively, *p* = 0.022). In conclusion, there was an association of serum osmolarity with clinical status and both in-hospital and out-of-hospital outcomes. Moreover, the linear dependence between vasopressin concentration and serum osmolarity in the AHF population was identified and was driven mainly by patients with de novo AHF which suggests different pathophysiological paths in ADHF and AHF de novo.

## 1. Introduction

Acute heart failure (AHF) is a life-threatening entity with 25–30% one-year mortality and a very high risk of rehospitalization [1]. One of its clinical manifestations is congestion, which occurs in 80% of cases on admission to the hospital. One of the principal reasons for AHF episodes is the inability to control euvolemia and proper water–electrolyte balance [2,3,4]. Its pathogenesis contributes to several factors, such as a natriuretic response, volume displacement, and neurohormonal activity [4,5,6]. Vasopressin is an important neurohormonal factor connecting kidney function and the body’s water accumulation. It is called the antidiuretic hormone because, in addition to activities such as vasoconstriction it causes water reabsorption by acting in the renal collecting tubule [7,8,9]. Its antidiuretic effect may result in fluid retention and deterioration of HF [10].

Osmolarity is an essential factor in water–electrolyte homeostasis maintenance. It depends on several well-known factors such as sodium, urea, and glucose [11]. The serum osmolarity reference range is relatively narrow despite its broader concentration norms. This means that osmolyte concentration changes should be immediately compensated, especially by free-water volume. Therefore, serum osmolarity may, to some degree, reflect total body fluid volume [7]. Proper osmolarity level maintenance mechanisms are, among others, the action of vasopressin and the Renin–Angiotensin–Aldosterone System (RAAS), which leads to an increase in the body’s neurohormonal activity—one of the well-known AHF pathways [12,13,14]. The most important organ for their effective action is the kidney. The pathophysiological interplay between the heart and kidney in AHF is called cardiorenal syndrome and resulting in renal dysfunction [15,16,17]. On the other hand, the kidneys are the key organs in the decongestion process, indirectly expressed by proper quantitative and qualitative urine excretion, which is dependent on vasopressin as a result of its tightly intertwined pathophysiology [4,9,18]. However, despite these facts, vasopressin and osmolarity are not commonly used markers in routine clinical practice. In contrast to RAAS, the role of serum osmolarity and vasopressin in the prognosis of AHF is still unclear.

This study aimed to evaluate the differences between highlighted serum osmolarity quartiles, especially in clinical features, laboratory parameters, kidney function, urine composition, and vasopressin concentration. We also wanted to establish the link between osmolarity, vasopressin, and outcomes in the AHF group, including de novo AHF and the acute decompensated acute heart failure (ADHF) subpopulations.

## 2. Material and Methods

This is a single-center, observational study. The study population included patients admitted with AHF to the Centre of Heart Diseases, 4th Military Hospital, Wroclaw, Poland. All participants were enrolled in the AHF registry carried out in 2010–2012 and 2016–2017. The inclusion criteria were as follows: AHF as the primary cause of hospitalization (according to the European Society of Cardiology HF Guidelines [19,20]), adults (age ≥ 18 years old), and a written informed consent provided by the patient. Exclusion criteria were cardiogenic shock, diagnosis of acute coronary syndrome, known severe liver disease, end-stage renal failure requiring renal replacement therapy, and evidence of infection. The study was approved by the local ethics committee (the Ethics Committee of Wroclaw Medical University) and was conducted in accordance with the Declaration of Helsinki and Good Clinical Practice.

### 2.1. Study Design

After admission to the hospital, detailed information related to participants’ demographic medical history, previous treatment, and co-morbidities was recorded. Clinical and laboratory examinations were carried out at four timepoints: admission, on the first and second day of hospitalization, and discharge. Participants were subjected to a 365-day follow-up.

### 2.2. Laboratory Parameters

Laboratory parameters such as N-terminal pro-B-type natriuretic peptide (NT-proBNP), troponin I, peripheral blood morphology, renal function parameters (urea, creatinine), electrolytes (sodium, potassium), liver function parameters (bilirubin, aspartate aminotransferase (AST), alanine aminotransferase (ALT), albumin), C-reactive protein (CRP), lactates in capillary blood, and urine spot samples (sodium, creatinine, urea) were performed in a hospital laboratory in standard laboratory tests during patient hospitalization. Vasopressin concentration was measured post hoc from frozen samples using the ELISA test.

### 2.3. Serum Osmolarity Determination

Baseline serum osmolarity was calculated from serum sodium, serum glucose, and urea using the equation: 1.86 × sodium [mmol/L] + (glucose [mg/dL]/18) + (urea [mg/dL]/2.8) + 9).

### 2.4. Subsection

The study population was divided into four equal quartiles based on a baseline serum osmolarity and marked as:low: <287 mOsm/L,intermediate low: 287–294 mOsm/L,intermediate high: 295–304 mOsm/L,high: >304 mOsm/L.

The paper presents all comparisons as low to high osmolarity quartiles, respectively.

Study endpoints were defined as:one-year all-cause mortality,worsening of HF (WHF),worsening of renal function (WRF).

The worsening of HF was defined as no improvement or deterioration of the patient’s condition during hospitalization [21,22]. The WRF was defined as an increase > 0.3 mg/dL of creatinine concentration within 48 h after admission [23].

### 2.5. Statistical Analysis

Continuous variables with normal distribution were presented by mean ± standard deviation (SD). Skewed variables were defined as a median [lower and higher quartile]. Categorized values were presented as numbers and/or percentages. Shapiro–Wilk tests were performed to check the normality of the distribution. The Levene test was used to evaluate homogeneity of variance.

Analysis of variance (ANOVA), Kruskal–Wallis tests with post-hoc tests were used to test for differences between quartiles in quantitative variables and the Chi-squared test was utilized for qualitative variables. The Mann–Whitney U-test, *t*-test, and Chi-squared test were applied to compare variables between two groups when appropriate (HF de novo and Acute Decompensated Heart Failure). Correlations between variables were assessed by the Pearson linear correlation test and Spearman correlation test. Kaplan–Meier curves were constructed to estimate the survival probability. The Cox Proportional Hazard Model was used to evaluate the prognostic significance of variables and its hazard ratio to one year mortality. The multivariable regression model was made to establish independent factors that influenced osmolarity. Those *p* values < 0.05 were considered statistically significant. Analysis was performed with STATISTICA 12 software (StatSoft Polska Sp. z o.o., Kraków, Poland).

## 3. Results

### 3.1. Baseline Study Group Characteristics

The study population included 361 patients, predominantly male (76.3%), with a mean age of 68 ± 13 years. The mean left ventricle ejection fraction (LVEF) was 34 ± 13%. The majority of patients (242; 67%) had decompensation of chronic HF. The median of NTproBNP was 5567 [3189–10382] pg/mL.

Among our population, we were able to calculate osmolarity in 338 patients (93.6%). Its mean value was 296 ± 14 mOsm/L, while the median vasopressin concentration was 54.3 [38.0–78.6] pg/mL.

### 3.2. Comparison of Baseline Characteristics by Osmolarity Quartiles

There were significant differences in patient age, which increased across the osmolarity quartile (from low to high): (63 ± 14 vs. 66 ± 13 vs. 69 ± 12 vs. 74 ± 11 years, respectively, *p* < 0.001). There was a difference in systolic blood pressure between the osmolarity quartiles (122 ± 27 vs. 135 ± 29 vs. 140 ± 35 vs. 133 ± 31 mmHg, respectively, *p* = 0.002), and the median heart rate was similar in all groups (90 [75–105] vs. 90 [75–110] vs. 80 [71–100] vs. 80 [70–100] beat per minute, respectively, *p* = 0.090).

Patients from the high osmolarity group stayed in the hospital for a more extended period (7 [5–10] vs. 6 [5–8] vs. 7 [5–10] vs. 8 [6–12] days, respectively, *p* = 0.003). Additionally, they had more co-morbidities such as arterial hypertension, chronic kidney disease, and diabetes mellitus (60 vs. 73 vs. 80 vs. 80%, respectively, *p* = 0.015 and 35 vs. 37 vs. 60 vs. 80%; *p* < 0.001 and 29 vs. 33 vs. 38 vs. 55%, respectively, *p* = 0.034) (Table 1.).

The inotropic agents were administered significantly more frequently in patients belonging to the low and high osmolarity quartiles (14 vs. 4 vs. 8 vs. 17%, respectively, *p* = 0.026). There was no difference in the use of other guideline-recommended pharmacotherapy such as angiotensin-converting enzyme (ACE) inhibitors and angiotensin receptor blockers (ARB), mineralocorticoid receptor antagonists (MRA), beta-blockers, loop diuretics, or nitroglycerine between the groups (Table 2).

### 3.3. Basic Laboratory Parameters by Osmolarity Quartiles

The decreasing bilirubin concentration was observed across quartiles (1.5 [0.9–2.2] vs. 1.0 [0.7–1.5] vs. 0.9 [0.7–1.5] vs. 1.1 [0.7–1.7] mg/dL, respectively, *p* < 0.001), with no differences in AST and ALT. Albumin concentration was lower in the low osmolarity quartile compared to the rest of the population (3.6 ± 0.4 vs. 3.8 ± 0.4 vs. 3.8 ± 0.4 vs. 3.7 ± 0.3 mg/dL, respectively, *p* = 0.043).

Median NT-proBNP and lactate concentration had a U-shaped distribution; thus, the highest values were observed in extreme quartiles—low and high (6124 [3189–11958] vs. 4191 [2608–7550] vs. 5363 [2671–9930] vs. 7611 [5255–8654] pg/mL, respectively, *p* < 0.001 for NT-proBNP and 2.2 [1.8–2.7] vs. 1.8 [1.5–2.2] vs. 1.6 [1.4–2.2] vs. 2.0 [1.5–2.5] mmol/L, respectively, *p* = 0.001, for lactate) (Table 1).

### 3.4. Kidney Function Parameters and Components of Serum Osmolarity by Quartiles

Kidney function parameters such as urea and creatinine increased gradually across osmolarity quartiles (37 [28–48] vs. 44 [37–54] vs. 56 [43–68] vs. 89 [71–115] mg/dL, respectively, p < 0.001, and 1.0 ± 0.3 vs. 1.1 ± 0.3 vs. 1.3 ± 0.4 vs. 1.9 ± 0.8 mg/dL, respectively, p < 0.001). The same trend was observed in the case of sodium and potassium concentration (135 [133–138] vs. 139 [138–141] vs. 141 [138–143] vs. 141 [138–144] mmol/L, respectively, *p* < 0.001, and 3.9 [3.6–4.4] vs. 4.2 [3.8–4.5] vs. 4.3 [3.9–4.5] vs. 4.2 [4.0–4.8] mmol/L, respectively, *p* = 0.002). The highest glucose concentration was recorded in the high serum osmolarity quartile (118 ± 31 vs. 126 ± 47 vs. 139 ± 56 vs. 152 ± 74 mg/dL, *p* = 0.002) (Table 1).

### 3.5. Urine Composition by Osmolarity Quartiles

The lowest spot urine sodium was observed in the group with the lowest serum osmolarity (53 [32–88] vs. 84 [39–110] vs. 73 [49–113] vs. 86 [49–118] mmol/L, respectively, *p* = 0.040). Distribution of urine urea concentration was U-shaped, with the lowest values in extreme quartiles (1128 [698–1788] vs. 1441 [1151–2076] vs. 1211 [593–1681] vs. 986 [594–1401] mg/dL, respectively, *p* = 0.010). Urine creatinine concentration decreased gradually along with quartiles (122 [39–175] vs. 128 [74–217] vs. 84 [42–131] vs. 65 [35–98] mg/dL, respectively, *p* = 0.006) (Table 1).

No correlation was found between urine sodium, creatinine, and osmolarity or vasopressin concentration.

### 3.6. The Relationship between Serum Osmolarity and Vasopressin Concentration

Vasopressin concentration decreased significantly by higher osmolarity quartile (61.6 [44.0–81.0] vs. 57.8 [50.0–77.3] vs. 52.7 [43.1–69.2] vs. 45.0 [30.7–60.7] pg/mL, respectively, *p* = 0.034) (Table 1) and presented linear correlation with osmolarity (r = −0.221, *p* = 0.003). Other factors which had an influence on osmolarity—patients’ age and serum potassium concentration—were established in multivariate analysis (Appendix A).

### 3.7. The Comparison of De Novo and Acute Decompensated Chronic Heart Failure Regarding Vasopressin Concentration, Kidney Function, and Urine Laboratory Parameters by Osmolarity

There was a significant difference in vasopressin concentration between the quartiles among de novo AHF (63.6 [55.3–94.5] vs. 58.0 [50.7–78.6] vs. 52.0 [46.0–58.0] vs. 38.0 [27.0–57.0] pg/mL, respectively, *p* = 0.022). In contrast, no differences were observed in vasopressin between the quartiles in ADHF (59.5 [37.4–80.0] vs. 52.0 [38.0–74.5] vs. 57.0 [38.0–79.0] vs. 50.0 [33.0–84.0] pg/mL, respectively, *p* = 0.849). No difference in the osmolarity and vasopressin concentration between HF de novo and ADHF was found (297 ± 13 vs. 296 ± 15 mOsm/L, respectively, *p* = 0.451, and 55.0 [41.5–70.4] vs. 52.7 [35.0–81.0] pg/mL, respectively, *p* = 0.702).

Urinary sodium levels tended to increase with osmolarity quartiles in the HF de novo group with a borderline *p*-value (58 [37–85] vs. 77 [38–98] vs. 73 [53–108] vs. 108 [75–134] mmol/L, respectively, *p* = 0.053). This was not observed in the ADHF subpopulation (52 [32–88] vs. 89 [40–113] vs. 62 [38–118] vs. 73 [47–94] mmol/L, respectively, *p* = 0.472).

Urea and creatinine concentrations in urine showed a U-shaped distribution with the lowest values in the group with the low and high osmolarity in de novo HF patients (1560 [1045–2000] vs. 1796 [1207–2533] vs. 1173 [650–1681] vs. 1014 [658–1362] mg/dL, respectively, *p* = 0.011, and 141 [51–261] vs. 164 [78–240] vs. 95 [60–124] vs. 55 [41–114] mg/dL, respectively, *p* = 0.036), without significant differences in ADHF patients (1029 [689–1577] vs. 1013 [758–1787] vs. 1492 [744–1650] vs. 960 [594–1466] mg/dL, respectively, *p* = 0.376, and 113 [38–145] vs. 88 [39–151] vs. 86 [42–132] vs. 64 [34–92] mg/dL, respectively, *p* = 0.190). (Table 3).

### 3.8. Outcomes of the Study

There was a statistically significant, U-shaped distribution of one-year mortality and WHF with the highest number of episodes in the low and the high osmolarity groups (31% vs. 19% vs. 23% vs. 37%; respectively, *p* = 0.022, and 20% vs. 9% vs. 10% vs. 22%, respectively, *p* = 0.032) (Figure 1, Table 4). Kaplan-Meier curves presents survival probability by quartiles for one-year follow-up (Figure 2). Despite this, osmolarity and vasopressin were not predictive of death at one-year observation (HR: 1.01 (0.99–1.02), *p* = 0.158 and HR: 1.00 (0.99–1.01), *p* = 0.828, respectively). The relationships between serum osmolarity, vasopressin, and Hazard Ratio (HR) regarding the outcome were shown in the supplement (Appendix A).

WRF incidents were more frequent in patients with higher osmolarity (in the intermediate high and high osmolarity quartile) (4% vs. 2% vs. 13% vs. 11%; *p* = 0.018, respectively) (Figure 1, Table 4).

## 4. Discussion

The issue of osmolarity is of great interest in the context of heart failure. Our study provides insight into the understanding of serum osmolarity and its neurohormonal (vasopressin) control in AHF pathophysiology. First, the relationship between serum osmolarity and one-year mortality and WHF in the study population was not linear. The highest risk of mortality and worsening HF was observed in patients in the low and the high quartile of serum osmolarity. The risk, therefore, had a U-shape trajectory. It is essential to acknowledge that patients with low serum osmolarity, despite being relatively young, exhibited characteristics of more severe HF. These included: lower blood pressure and lower serum sodium concentration as well as: high NTproBNP, elevated bilirubin, and low albumin concomitant with the highest vasopressin concentration [24,25]. On the other hand, the high osmolarity group had a different profile. They were the oldest, with the greatest number of comorbidities and renal dysfunction probably being contributors to poor outcomes in high osmolarity. The results show two distinct phenotypes of AHF patients, both with poor outcomes but most likely resulting from diverse mechanisms. The research of Nakagawa et al. gives interesting results, where it was observed that the highest serum osmolarity is associated with the highest mortality. It was, however, a study conducted in a group of patients with heart failure with a preserved left ventricle ejection fraction (HFpEF). Therefore, the characteristics of the study population were also specific. The female gender was dominant and the mean age was higher than in our study group. However, the common feature was higher comorbidity: chronic kidney disease, diabetes, and arterial hypertension in patients with high osmolarity, which were probably some of the essential factors accounting for a worse prognosis [14].

The low osmolarity group exhibited high sodium avidity, with low urine sodium excretion and high urine creatinine concentration. On the other hand, the high serum osmolarity quartile had high sodium excretion but low eGFR. This further confirms glomerular and tubular dysfunction dissociation in AHF, which may be associated with a worse decongestion process and poor diuretic response leading to HF worsening during hospitalization [26,27,28]. Additionally, both extreme profiles had elevated serum lactate, which was already shown to be a strong prognosticator in AHF [29,30,31].

There was a significant correlation between serum osmolarity and vasopressin in the population. However, the latter was not confirmed in the subgroup analysis of the de novo and ADHF patients. Interestingly, there was no gradient of vasopressin across the serum osmolarity quartiles in patients with ADHF. All quartiles had the same vasopressin level (~55 pg/mL). These findings might indicate that patients with acutely decompensated HF had relatively elevated vasopressin independent from serum osmolarity (so-called non-osmotic release of the hormone), which may have an influence for worse diuretic response and in-hospital course. On the other hand, there was a strong correlation between vasopressin and serum osmolarity in patients with de novo AHF. This concept refers to the gradual activation and dysfunction of several pathophysiological mechanisms [28].

Vasopressin is responsible for the regulation of serum osmolarity. Acting in the collecting tubule and the distal part of the ascending tubule, it induces expression of aquaporin, thereby increasing osmotic reabsorption of water. Moreover, it activates sodium transporters in the ascending tubule of the nephron, allowing iso-osmolar transport of sodium ions from the lumen of the tubule to the renal interstitium [7]. Through V2 receptors, it increases the activity of urea transporters, facilitating its reabsorption. The aforementioned mechanisms result in water reabsorption and urine concentration. Thus, the serum osmolarity seems to be an important factor in AHF pathophysiology.

These results contribute to a better understanding of the pathophysiology of AHF. This is extremely important in more effectively identifying groups with a high risk of a worse clinical course in patients with AHF already at the initial stage of inpatient treatment. We strongly believe that in the future they will contribute to the development of new diagnostic and therapeutic strategies aimed at improving the prognosis of patients with AHF.

## 5. Limitations

This is a retrospective single-center study with analysis and division of patients performed post hoc. The number of participants was relatively low. There is also limited information regarding diuresis, which may be a key to a better understanding of the process. Osmolarity was calculated from a mathematical formula based on the concentration of osmolytes. The method for precisely measuring osmolality is osmometry. There is a difference between osmolality and osmolarity; however, it is small, so it allows the use of mathematical formula to make an estimation [11]. Given the variability of the factors from which the osmolarity was calculated, it seems reasonable to conduct a similar study with its measured values.

## 6. Conclusions

Serum osmolarity is associated with both short-term and long-term outcomes in AHF patients (however, this relationship is not linear). Consequently, its implementation could potentially facilitate risk stratification of AHF patients. Further research is required to accurately assess the predictive role of this parameter.

## Figures and Tables

**Figure 1 biomedicines-10-02034-f001:**
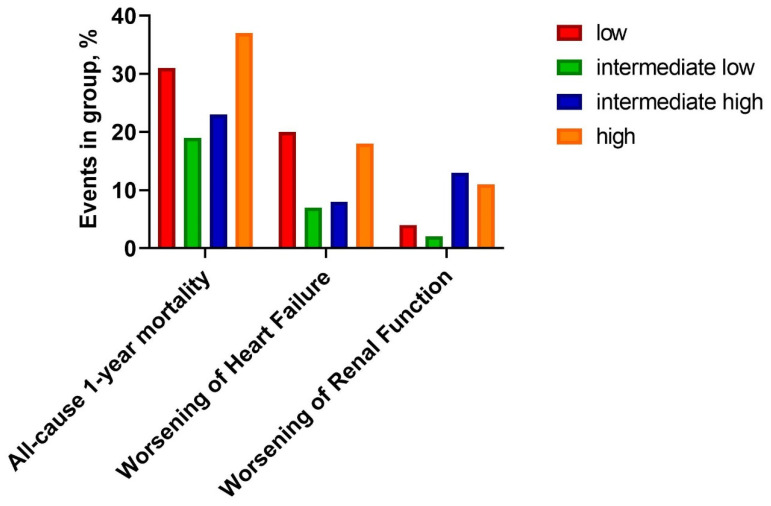
Clinical outcomes by quartiles.

**Figure 2 biomedicines-10-02034-f002:**
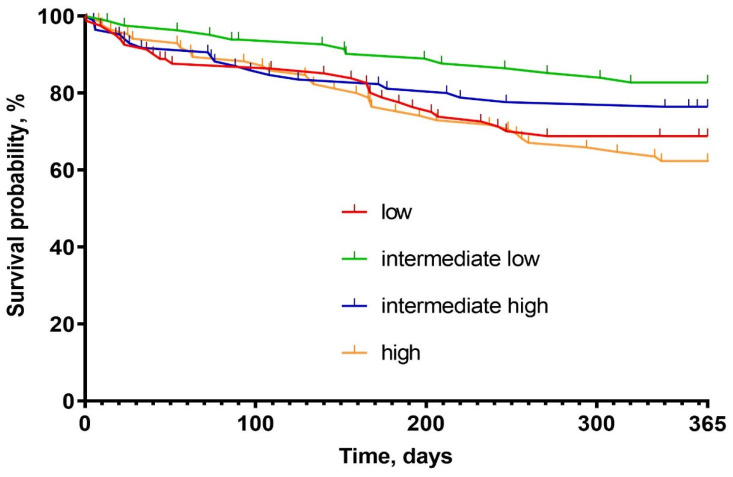
Kaplan–Meier survival probability graph.

**Table 1 biomedicines-10-02034-t001:** Comparison of clinical and laboratory parameters between quartiles.

Serum Osmolarity, mOsm/L	Low	Intermediate Low	Intermediate High	High	*p*-Value
Age, years	63 ± 14	66 ± 13	69 ± 12	74 ± 11	<0.001
Men, N (%)	61 (74)	63 (76)	63 (74)	71 (81)	0.780
Systolic BP, mmHg	122 ± 27	135 ± 29	140 ± 35	133 ± 31	0.002
Diastolic BP, mmHg	74 ± 13	79 ± 15	81 ± 18	80 ± 18	0.043
Heart rate, b.p.m	90 [75–105]	90 [75–110]	80 [71–100]	80 [70–100]	0.090
Body weight, kg	80 ± 20	82 ± 18	82 ± 18	82 ± 15	0.913
LVEF, %	30 [20–43]	30 [25–45]	31 [25–40]	35 [23–48]	0.804
AHF de novo, N (%)	20 (24)	32 (40)	32 (39)	23 (27)	0.073
Arterial hypertension, N (%)	49 (60)	60 (73)	67 (80)	68 (80)	0.015
Chronic kidney disease, N (%)	26 (35)	29 (37)	46 (60)	69 (80)	<0.001
Diabetes mellitus, N (%)	23 (29)	26 (33)	32 (38)	48 (55)	0.034
Hospitalization length, days	7 [5–10]	6 [5–8]	7 [5–10]	8 [6–12]	0.003
AST, IU/L	28 [23–40]	26 [20–36]	28 [21–38]	26 [18–38]	0.465
ALT, IU/L	29 [21–43]	28 [20–52]	29 [18–46]	25 [16–45]	0.512
Bilirubin, mg/dL	1.5 [0.9–2.2]	1.0 [0.7–1.5]	0.9 [0.7–1.5]	1.1 [0.7–1.7]	<0.001
Albumin, mg/dL	3.6 ± 0.4	3.8 ± 0.4	3.8 ± 0.4	3.7 ± 0.3	0.043
Hemoglobin, g/dL	13 ± 2	13 ± 2	13 ± 2	12 ± 2	0.001
CRP, mg/L	10 [5–22]	6 [3–14]	7 [3–14]	6 [3–17]	0.134
Glucose, mg/dL	118 ± 31	126 ± 47	139 ± 56	152 ± 74	0.002
Creatinine, mg/dL	1.0 ± 0.3	1.1 ± 0.3	1.3 ± 0.4	1.9 ± 0.8	<0.001
Urea, mg/dL	37 [28–48]	44 [37–54]	56 [43–68]	89 [71–115]	<0.001
Na^+^, mmol/L	135 [133–138]	139 [138–141]	141 [138–143]	141 [138–144]	<0.001
K^+^, mmol/L	3.9 [3.6–4.4]	4.2 [3.8–4.5]	4.3 [3.9–4.5]	4.2 [4.0–4.8]	0.002
Lactate, mmol/L	2.2 [1.8–2.7]	1.8 [1.5–2.2]	1.6 [1.4–2.2]	2.0 [1.5–2.5]	0.001
NT-proBNP, pg/mL	6124 [3189–11958]	4191 [2608–7550]	5363 [2671–9930]	7611 [5255–8654]	<0.001
Vasopressin, pg/mL	61.6 [44.0–81.0]	57.8 [50.0–77.3]	52.7 [43.1–69.2]	45.0 [30.7–60.7]	0.034
Urine sodium, mmol/L	53 [32–88]	84 [39–110]	73 [49–113]	86 [49–118]	0.040
Urine urea, mg/dL	1128 [698–1788]	1441 [1151–2076]	1211 [593–1681]	986 [594–1401]	0.010
Urine creatinine, mg/dL	122 [39–175]	128 [74–217]	84 [42–131]	65 [35–98]	0.006

**Table 2 biomedicines-10-02034-t002:** Comparison of Heart Failure treatment between quartiles.

Serum Osmolarity, mOsm/kg	Low	Intermediate Low	Intermediate High	High	*p*-Value
Loop diuretics, N (%)	85 (100)	80 (96)	84 (99)	85 (100)	0.101
Inotropes, N (%)	12 (14)	3 (4)	7 (8)	14 (17)	0.026
Beta-blocker, N (%)	81 (96)	80 (96)	84 (99)	77 (92)	0.128
ACEI/ARB, N (%)	72 (87)	72 (89)	76 (94)	72 (86)	0.361
MRA, N (%)	42 (51)	41 (51)	29 (46)	36 (53)	0.155

**Table 3 biomedicines-10-02034-t003:** Comparison of de novo and acute decompensated heart failure.

Serum Osmolarity, mOsm/L		Low	Intermediate Low	Intermediate High	High	*p*-Value
Vasopressin, pg/mL	AHF de novo	63.6 [55.3–94.5]	58.0 [50.7–78.6]	52.0 [46.0–58.0]	38.0 [27.0–57.0]	0.002
	ADHF	59.5 [37.4–80.0]	52.0 [38.0–74.5]	57.0 [38.0–79.0]	50.0 [33.0–84.0]	0.849
Urine sodium, mmol/L	AHF de novo	58 [37–85]	77 [38–98]	73 [53–108]	108 [75–134]	0.053
	ADHF	52 [32–88]	89 [40–113]	62 [38–118]	73 [47–94]	0.472
Urine creatinine, mg/dL	AHF de novo	141 [51–261]	164 [78–240]	95 [60–124]	55 [41–114]	0.036
	ADHF	113 [38–145]	88 [39–151]	86 [42–132]	64 [34–92]	0.190
Urine urea, mg/dL	AHF de novo	1560 [1045–2000]	1796 [1207–2533]	1173 [650–1681]	1014 [658–1362]	0.011
	ADHF	1029 [689–1577]	1013 [758–1787]	1492 [744–1650]	960 [594–1466]	0.376
Serum sodium, mmol/L	AHF de novo	137 [136–139]	140 [139–141]	141 [140–144]	143 [140–146]	<0.001
	ADHF	135 [131–138]	138 [135–141]	140 [137–143]	141 [137–143]	<0.001
Serum creatinine, mg/dL	AHF de novo	0.93 [0.81–1.20]	1.02 [0.88–1.10]	1.15 [1.01–1.39]	1.70 [1.34–1.96]	<0.001
	ADHF	0.96 [0.85–1.20]	1.23 [1.05–1.43]	1.26 [1.06–1.51]	1.73 [1.38–2.12]	<0.001
Serum urea, mg/dL	AHF de novo	35 [27–38]	40 [33–48]	45 [40–57]	79 [58–105]	<0.001
	ADHF	39 [31–51]	44 [39–62]	58 [49–71]	89 [70–119]	<0.001

**Table 4 biomedicines-10-02034-t004:** Comparison of outcomes between quartiles.

Serum Osmolarity, mOsm/L	Low	Intermediate Low	Intermediate High	High	*p*-Value
All-cause one-year mortality, N (%)	25 (31)	14 (19)	20 (23)	32 (37)	0.022
Worsening of Heart Failure, N (%)	16 (20)	7 (9)	8 (10)	18 (22)	0.032
Worsening of Renal Function, N (%)	3 (4)	2 (2)	11 (13)	10 (11)	0.018

## Data Availability

Not applicable.

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
