# Peer review of "Serum Osmolarity and Vasopressin Concentration in Acute Heart Failure—Influence on Clinical Course and Outcome"

_biomedicines, 2022, doi:10.3390/biomedicines10082034_

Round 1

Reviewer 1 Report

The manuscript by Guzik et. al has shown the relation between serum osmolarity and vasopressin concentration in the heart failure patients. Although the topic of the manuscript is interesting but the study fails to provide a solid conclusion on effects of  serum osmolarity and vasopressin concentration in worsening of heart failure. I have following comments for the manuscript:

1. The abstract is very complicated and hard to follow. Please improve .

2. The manuscript objectives are not clear, introduction needs to be rewritten to show the importance of osmolarity/vasopressin concentration in heart failure patients with proper citations.

3. Results are confusing and do not elaborate the outcome.

4. Please provide a paragraph on how this study will benefit the clinical decision making/translational outlook.

Author Response

Dear Sir or Madam,
thank you very much for all comments regarding the article.
The detailed replies to your review are included in the file attached below.

Yours sincerely,

Mateusz Guzik

Reviewer 2 Report

The present study demonstrated that U-shape relationship between serum osmolarity and outcome and confirmed the negative relationship between serum osmolarity and vasopressin levels in acute heart failure. U-shape relationship between serum osmolarity and outcome is interesting, but there is a study showed that the higher osmolarity, the higher mortality ( BMC Cardiovasc Disord (2021) 21:281). What are independent factors that determine serum osmolarity? Serum osmolarity must be influenced by several factors except vasopressin. Please provide a figure which shows the relationships between serum osmolarity and hazard ratio (HR) regarding outcome and also the relationship between vasopressin levels and HR. 

Author Response

Dear Sir or Madam,

I am grateful for any comments that undoubtedly allowed to improve the quality of the manuscript.

Detailed replies to the comments are included in the file attached below.

Your sincerely,

Mateusz Guzik

Round 2

Reviewer 1 Report

The manuscript has been improved after the incorporation of suggestions. I have no further comments.

Author Response

Dear Sir / Madam,

thank you for reviewing our article. Undoubtedly, it contributed to the improvement of its quality. We are grateful for all the valuable comments and the final positive reception of our work.

Yours faithfully,

Mateusz Guzik
